# Unimolecular net heterolysis of symmetric and homopolar σ-bonds

Anna F. Tiefel[1,8], Daniel J. Grenda[2,8], Carina Allacher[2], Elias Harrer[1], Carolin H. Nagel[1], Roger J. Kutta[2], David Hernández-Castillo[3,4], Poorva R. Narasimhamurthy[1], Kirsten Zeitler[5], Leticia González[3,6], Julia Rehbein[1✉], Patrick Nuernberger[2,7✉] & Alexander Breder[1✉]

The unimolecular heterolysis of covalent σ-bonds is integral to many chemical transformations, including $S_N1$-, E1- and 1,2-migration reactions. To a first approximation, the unequal redistribution of electron density during bond heterolysis is governed by the difference in polarity of the two departing bonding partners[1–3]. This means that if a σ-bond consists of two identical groups (that is, symmetric σ-bonds), its unimolecular fission from the $S_0$, $S_1$, or $T_1$ states only occurs homolytically after thermal or photochemical activation[1–7]. To force symmetric σ-bonds into heterolytic manifolds, co-activation by bimolecular noncovalent interactions is necessary[4]. These tactics are only applicable to σ-bond constituents susceptible to such polarizing effects, and often suffer from inefficient chemoselectivity in polyfunctional molecules. Here we report the net heterolysis of symmetric and homopolar σ-bonds (that is, those with similar electronegativity and equal leaving group ability[3]) by means of stimulated doublet–doublet electron transfer (SDET). As exemplified by Se–Se and C–Se σ-bonds, symmetric and homopolar bonds initially undergo thermal homolysis, followed by photochemically SDET, eventually leading to net heterolysis. Two key factors make this process feasible and synthetically valuable: (1) photoexcitation probably occurs in only one of the incipient radical pair members, thus leading to coincidental symmetry breaking[8] and consequently net heterolysis even of symmetric σ-bonds. (2) If non-identical radicals are formed, each radical may be excited at different wavelengths, thus rendering the net heterolysis highly chemospecific and orthogonal to conventional heterolyses. This feature is demonstrated in a series of atypical $S_N1$ reactions, in which selenides show SDET-induced nucleofugalities[3] rivalling those of more electronegative halides or diazoniums.

According to valence bond theory, the fission of σ-bonds from their ground ($S_0$) or lowest excited states ($S_1$, $T_1$) through a single elementary step cannot be heterolytic in nature if their constituents are identical (that is, symmetric σ-bonds, Fig. 1)[1,2]. This exclusion criterion[1] has profoundly shaped scientists' notion of σ-bonds as being amenable or unamenable to single-step heterolysis (Fig. 1a,b). In organic molecules, σ-bonds susceptible to unimolecular heterolysis are typically composed of carbon-bound heteroatoms (that is, heteronuclear σ-bonds) with distinct differences in electronegativity relative to carbon, showing, in part, large dipole moments along the carbon–element axis (that is, heteropolar σ-bonds)[2]. Until today, experimental studies on single-step thermal and photochemical heterolyses have only focused on heteropolar σ-bonds but left analogous reactions of symmetric and homopolar σ-bonds (meaning, bond constituents with similar electronegativity and equal fugality[3]) virtually uninvestigated. From a methodological viewpoint, this state of affairs is very

deplorable, because an immense synthetic potential may arise from the ionic fission of symmetric and homopolar σ-bonds. More concretely, supposing that such a polar cleavage requires an activation principle that is fundamentally different from conventional protocols (that is, single-step thermal or photochemical), it is expected that such a putative orthogonality enables σ-bond cleavage even in the presence of various other functionalities, and thus probably results in high regioselectivity and chemospecificity concomitant with elevated reactivity[9,10].

On the basis of these notions, we wondered whether symmetric σ-bonds may deviate from their natural tendency to solely undergo single-step homolysis (Fig. 1b) but instead engage in a two-step heterolytic pathway if the bond activation would consist of two separate stimuli (Fig. 1c). We posited that the first stimulus, for example, thermal excitation, would result in the expected homolysis to furnish a radical pair (for example, two doublets). Subsequent photoexcitation (second stimulus) of one of the incipient radicals at a proper wavelength

[1]Institut für Organische Chemie, Fakultät für Chemie und Pharmazie, Universität Regensburg, Regensburg, Germany. [2]Institut für Physikalische und Theoretische Chemie, Fakultät für Chemie und Pharmazie, Universität Regensburg, Regensburg, Germany. [3]Institute of Theoretical Chemistry, Faculty of Chemistry, University of Vienna, Vienna, Austria. [4]Doctoral School in Chemistry (DoSChem), University of Vienna, Vienna, Austria. [5]Fakultät für Chemie und Mineralogie, Universität Leipzig, Leipzig, Germany. [6]Vienna Research Platform on Accelerating Photoreaction Discovery, University of Vienna, Vienna, Austria. [7]Regensburg Center for Ultrafast Nanoscopy (RUN), University of Regensburg, Regensburg, Germany. [8]These authors contributed equally: Anna F. Tiefel, Daniel J. Grenda. ✉e-mail: julia.rehbein@ur.de; patrick.nuernberger@ur.de; alexander.breder@ur.de

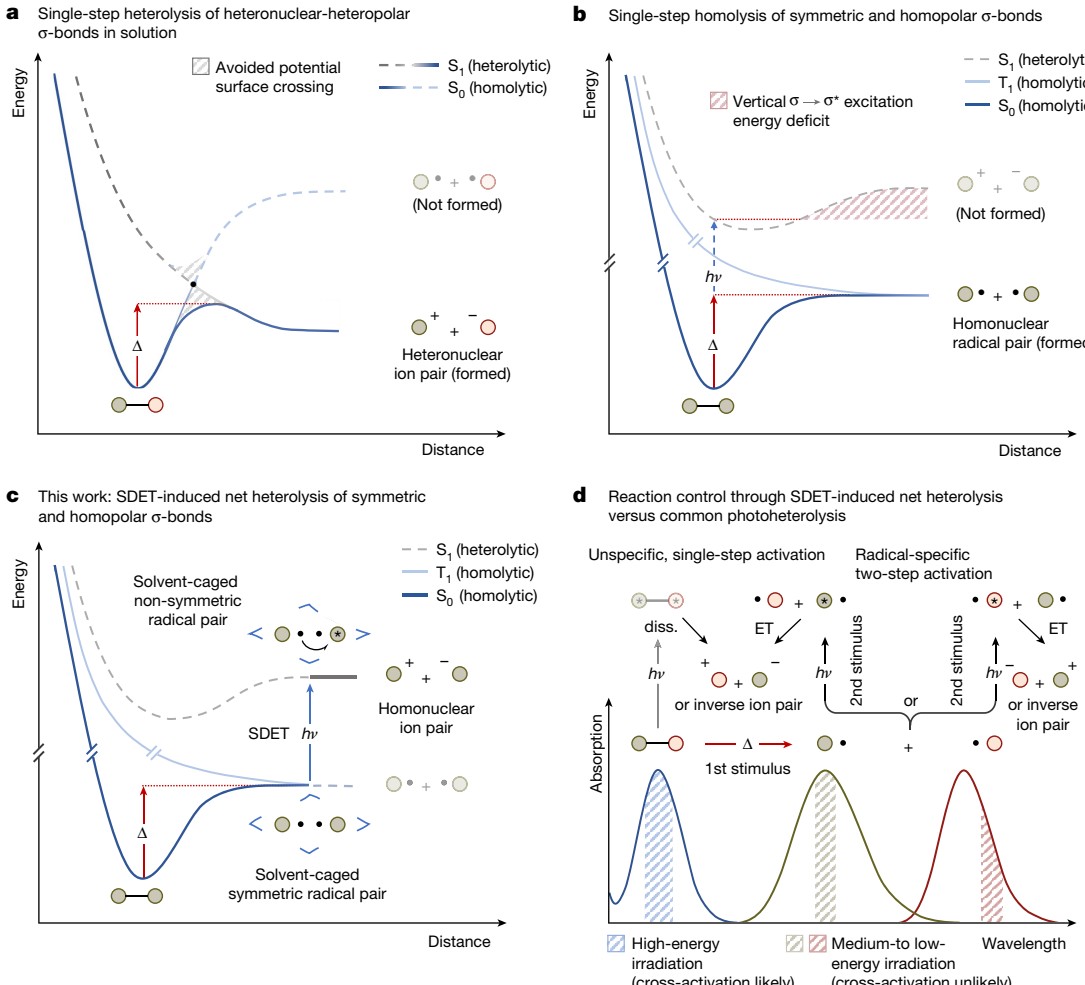

**Fig. 1 | Representation of potential energy surfaces for homo- and heteronuclear σ-bond fission and conceptualization of SDET-induced σ-bond net heterolysis. a**, Heteropolar σ-bond constitution leads to intersection of the $S_0$ and $S_1$ surfaces in solution, resulting in heterolysis by means of avoided surface crossing[2]. The $T_1$ surface is omitted for clarity. **b**, Thermal and photochemical single-step fissions of symmetric and homopolar σ-bonds exclusively proceed along the homolytic $S_0$ or $T_1$ surfaces. The vertical singlet σ → σ* excitation energy of symmetric σ-bonds is smaller than the

heterolytic σ-bond dissociation (diss.) energy[1] due to electric potential and solvent reorganization energies[5,6] needed to form ambipolar ion pairs[10]. **c**, Coincidental photoexcitation of one radical next to its homonuclear ground state radical partner inside a solvent cage leads to symmetry breaking[7] and elevation onto the heterolytic $S_1$ potential surface after electron transfer. **d**, Enhanced reaction control in the polar fission of σ-bonds by selective irradiation (green and bordeaux-coloured pathways) of incipient, σ-bond-derived radicals in contrast to direct σ-bond photolysis. ET, electron transfer.

(Fig. 1c,d) would result in a stimulated doublet–doublet electron transfer (SDET), and consequently in a net σ-bond heterolysis.

Mechanistically, this cleavage can be regarded as an ampholysis reaction due to the formation of a constitutionally identical pair of ions with opposite charges through a sequence featuring elements of both homolysis and heterolysis[11]. This scenario is remarkable, as related two-step heterolyses are only known to proceed with heteropolar σ-bonds[12,13]. In such cases, the direction of the electron transfer is determined by the nature of the heteropolar radical pair (that is, unidirectional electron transfer from the electropositive to the electronegative radical).

Given that the photoexcitation step is coincidental and only likely to occur in one of the incipient radical pair members at any given time, symmetry restrictions imposed by valence bond or molecular orbital theory on the single-step heterolysis of symmetric σ-bonds no longer apply[1,6,7], as the electron transfer occurs only after the homolysis (Fig. 1c,d). Mechanistically similar electron transfers induced by coincidental, photochemical symmetry breaking were also reported for pairs of identical closed-shell molecules[8,14], which lends plausibility to the postulate that sufficiently longevous radical pairs with

unequal electronic or vibrational configurations may show an analogous behaviour.

Projection of this mechanistic hypothesis to non-symmetric, yet homopolar carbon–element σ-bonds[2] as targets for the SDET activation is expected to give controlled access to reactive carbenium intermediates from substrates that are inherently non-electrophilic. Consequently, these intrinsically inert substrates can engage in reaction manifolds that are typically observed only with heteropolar analogues, for example, in $S_N1$ reactions (that is, atypical $S_N1$ reactions) and 1,2-additions. On the basis of these considerations, we present herein a detailed mechanistic and synthetic study on the SDET activation of symmetric and homopolar selenium–selenium and carbon–selenium σ-bonds, respectively. We have identified relevant radical intermediates and deciphered their light-driven interconversion into highly reactive ion pairs, providing a profound groundwork for the design of unprecedented $S_N1$ reactions at non-electrophilic carbon centres carrying arylselanes as SDET-controlled leaving groups.

To assess the feasibility of the proposed SDET-induced ampholysis of symmetric σ-bonds, organic diselanes were chosen as suitable substrates. Previous reports by Xu et al.[15] showed that both aliphatic and

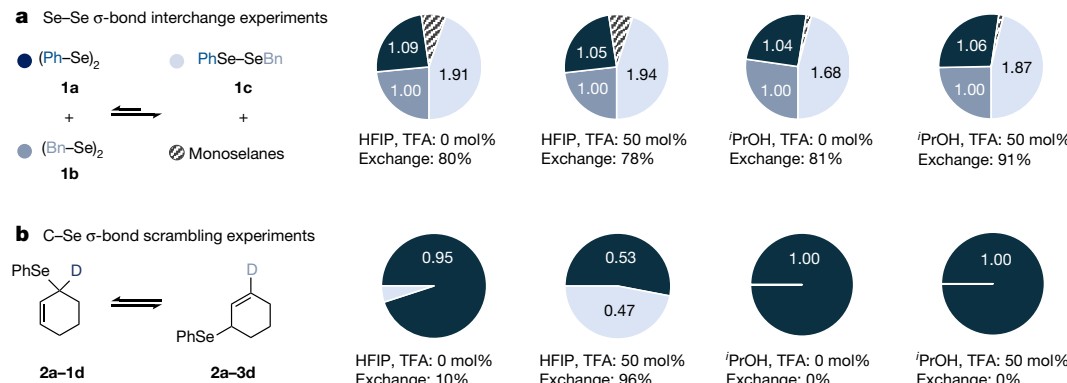

**Fig. 2 | Se–Se and C–Se σ-bond exchange experiments. a**, Composition of solutions initially containing a 1:1 mixture of **1a** and **1b** as starting materials at 19 °C in HFIP and $^i$PrOH. After 180 min, formation of **1c** and monoselanes is observed through Se–Se and Se–C σ-bond interchange, respectively. Numbers in the pie charts refer to relative amounts of **1a** (dark blue), **1b** (medium blue) and **1c** (light blue) in solution. **b**, Composition of solutions containing **2a–1d** as the starting material at 19 °C in HFIP and $^i$PrOH after 180 min.

aromatic diselanes readily undergo photochemical Se–Se σ-bond interchange through dynamic covalent reactions[16]. At room temperature and 5 mM concentration, the Se–Se σ-bond interchange of aliphatic diselanes was detected only under irradiation with ultraviolet or visible light[15]. This outcome was shown to be independent from all tested solvents (that is, chloroform, acetone, acetonitrile and methanol).

In a recent study from our laboratories on selenohydrins[17], we demonstrated that certain closed- and open-shell selenium species function as effective H-bond acceptors, and that fluorous alcoholic solvents such as 1,1,1,3,3,3-hexafluoropropan-2-ol (HFIP) have a stabilizing effect[17–21] on Se radical cations. To test whether akin solvent effects are operative with neutral Se radicals, equimolar solutions of diphenyldiselane (**1a**) and dibenzyldiselane (**1b**) were separately stirred in HFIP and propan-2-ol (19 °C, 0.5 M) in the dark. The degree of Se–Se σ-bond interchange (Fig. 2a) was monitored by $^1$H nuclear magnetic resonance (NMR) spectroscopy. After 3 h, corresponding interchange product **1c** was formed in each solvent with exchange percentages (that is, measured ratio of (**1c**:(**1a** + **1b**)) × 100% relative to the statistical ratio of (2:(1+1)) × 100%, Supplementary Table 16) ranging between 78 and 91%. In addition to **1c**, formation of benzylphenylselane and dibenzylselane was detected, which accounted for 2 to 9% of the consumed starting material. A similar outcome (85% Se–Se exchange) was observed when the σ-bond exchange was allowed to run for only 15 min in CDCl$_3$ in the dark, indicating that thermal Se–Se σ-bond homolysis is feasible[22] in protic and/or acidic solvents (p$K_a$ CHCl$_3$ = 13.6)[23] under experimental conditions. As expected, addition of 2,2,6,6-tetramethylpiperidinyl-1-oxyl (TEMPO) or galvinoxyl as radical scavengers markedly reduced the degree of Se–Se interchange (54 and 58%, respectively, Supplementary Table 16, entries 7 and 8). To test whether homopolar C–Se σ-bonds may also undergo homolysis under thermal conditions, 1-deuterocyclohex-2-en-1-yl(phenyl)selane (**2a–1d**) was stirred at 0.5 M concentration in both neat propan-2-ol and HFIP in the dark (Fig. 2b and Supplementary Table 15). No C–Se scrambling was observed in propan-2-ol in the course of 3 h. Addition of trifluoroacetic acid (TFA) (0.5 equiv.) to this solution did not alter this outcome, suggesting that Brønsted-acid-catalysed C–Se heterolysis does not play a decisive role under the tested conditions. Notably, performing C–Se scrambling in HFIP without acid additives resulted in 10% σ-bond interchange under otherwise unaltered conditions (Fig. 2b and Supplementary Table 15). Addition of 0.5 equiv. of TFA to **2a–1d** in HFIP led to 96% σ-bond interchange (that is, 48% of **2a–3d**) within less than 300 s in the dark. In the presence of TEMPO, the interchange was reduced to 78% and the formation of a TEMPO/cyclohex-2-en-1-yl recombination adduct was detected by electrospray ionization high resolution mass spectrometry (Supplementary

Information, page 15), which is congruent with a radical nature of the exchange reaction. In combination with our previous findings[17], these results strongly support our hypothesis on the stabilizing effect of fluorous alcoholic solvents on selenium radical intermediates. To substantiate this assumption, we quantified the stabilization energies exerted by HFIP and propan-2-ol on PhSe· and the cyclohexenyl radical by computation, using the B3LYP-D3/def2-TZVP//CPCM level of theory (Supplementary Table 21)[24–28]. These calculations show that HFIP indeed stabilizes both radical species slightly more effectively than propan-2-ol. Moreover, the increased chemical stability of the radicals after HFIP solvation can be rationalized by a substantial lowering of the singly occupied molecular orbital energies in both the allylic and the selenyl radical, resulting in a deceleration of the unproductive radical recombination (Supplementary Fig. 31). However, we interpret the benign impact of HFIP predominantly as a kinetic solvent effect[20].

As detailed in the Methods section, we showed that PhSe· ($\lambda_{max}$ = 490 nm) absorbs even above 500 nm, which is far more bathochromic than the absorptions of diselane **1a** itself, the selenium-centred ions resulting from the proposed SDET, and cyclohexenyl radical **5a·** (Fig. 3, Extended Data Fig. 1, Methods, and Supplementary Fig. 23). We therefore conducted next computations on the excited states of PhSe· (Supplementary Fig. 27) and its associated deactivation pathways (Supplementary Fig. 28) to determine the photochemically active states. Irradiation at wavelengths around 500 nm mainly populates the bright D$_3$ state (Supplementary Fig. 27a), which is of π$p$ excitation character (Supplementary Fig. 27b). The D$_2$ state is also of π$p$ excitation character but has a larger charge-transfer contribution (Supplementary Fig. 27b,c), which is consistent with its low extinction coefficient. Notably, in contrast to the D$_3$ and D$_2$ states, the D$_1$ state shows a n$_{Se}p$ charge-transfer character, which results in the localization of spin density in a $p$ orbital at the Se atom that is orthogonal to the adjacent π system of the phenyl ring (Supplementary Fig. 27d). Given the large excitation energy difference between the D$_2$ and the D$_1$ state (2.09 versus 0.43 eV), we assessed their deactivation pathways next to ascertain their potential contribution to the subsequent SDET process (Supplementary Fig. 28). To this end, we could determine a substantial activation barrier (0.83 eV) for the non-radiative deactivation of a vibrationally cooled D$_2$ state to the D$_1$ state, suggesting that at least a part of the excited PhSe· radicals may react from the D$_2$ state directly in an anti-Kasha manner[29,30].

On the basis of this analysis, we computed the standard absolute reduction potentials ($E^0$) of the D$_0$, D$_1$, and D$_2$ states of PhSe·, which amount to 4.33, 4.91, and 6.54 eV, respectively (Fig. 3a and Supplementary Information, Chapter 9). Considering the computationally estimated ground state reduction potential **5a$^+$**|**5a·** ($E^0$ = 4.75 eV), the

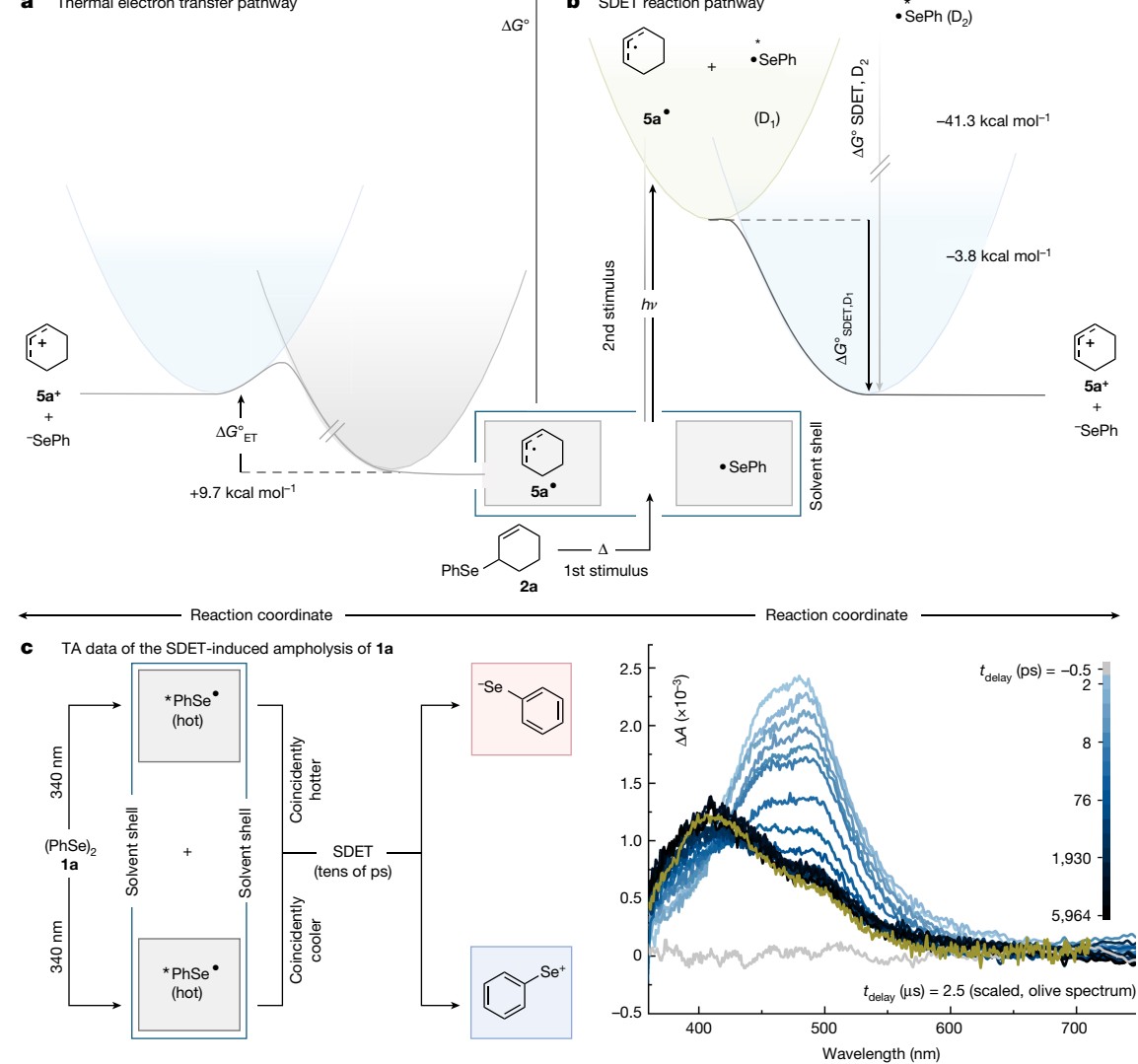

**a** Thermal electron transfer pathway

**b** SDET reaction pathway

**c** TA data of the SDET-induced ampholysis of **1a**

**Fig. 3 | Thermodynamic evaluation of a thermal electron transfer versus SDET between 5a˙ and ˙SePh, and transient absorption spectra of the SDET-induced ampholysis of 1a.** Potentials shown refer to reduction potentials, whereas the index indicates whether the respective species is oxidized (ox) or reduced (red) in the reaction. Gibbs free energies $\Delta G_{ET}$ and $\Delta G_{SDET}$ were calculated according to $\Delta G = -F (E^0_{red} - E^0_{ox})$. **a**, Thermal electron transfer is endergonic by 9.7 kcal mol⁻¹. **b**, SDET process is exergonic by −3.8 kcal mol⁻¹ from the D₁ state and −41.3 kcal mol⁻¹ from the D₂ state (Fig. 3b). Parabola for the D₂ state

is not shown, but the arrows representing the vertical excitation into the D₃ state and relaxation from the D₂ state are shown in grey. **c**, Transient absorption (TA) spectra after photoexcitation of a sample of (PhSe)₂ in HFIP with an ultrafast laser (central wavelength 340 nm, pulse duration 100 fs). The olive spectrum was recorded on a μs time scale after excitation at 355 nm and matches the end of the fs-transient absorption, indicating that the electron transfer stops after the end of vibrational cooling.

Gibbs free energy of an electron transfer ($\Delta G_{ET}$) from **5a˙** to PhSe˙ in their respective electronic ground states is endergonic by 9.7 kcal mol⁻¹ (that is, 0.42 eV, Fig. 3a), whereas after irradiation at 500 nm the corresponding SDET ($\Delta G_{SDET}$) becomes overall exergonic by −3.8 kcal mol⁻¹ from the D₁ state and −41.3 kcal mol⁻¹ from the D₂ state (Fig. 3b). These values suggest that allylic selanes such as **2a** can principally engage in atypical S_N1 reactions when exposed to nucleophiles under SDET conditions.

Eventually, two questions remained: (1) is it possible to obtain direct spectroscopic evidence for the SDET-induced Se−Se σ-bond ampholysis of **1a**, and can the resulting PhSe⁺/PhSe⁻ ion pair be intercepted by suitable reaction partners? (2) Can the archetypical reactivity profile of heteropolar carbon($sp^3$)−halogen σ-bonds (for example, halogen being Br) be emulated by SDET-induced net heterolysis of a homopolar carbon−selenium σ-bond (Fig. 4a), and can this concept indeed be used to conduct atypical S_N1 and 1,2-addition reactions (Fig. 4b, Extended Data Figs. 2 and 3)?

Regarding question (1) we learned that **1a** shows a signal of vibrationally excited (that is, hot) PhSe˙ arising at some point within 2 ps after excitation at 340 nm (Fig. 3c). Some tens of ps later, the spectrum turns into a double-peak structure, in which the rising absorption at 400 nm indicates the formation of PhSe⁺. Related photolyses of monoselanes do also show the formation PhSe˙ radicals but not that of PhSe⁺. We interpret this outcome as a result of a fast vibrational relaxation relative to diffusion. Meaning that if vibrationally relaxed (that is, cold) PhSe˙ radicals meet by diffusion, electron transfer is no longer energetically feasible. However, if vibrationally or electronically excited PhSe˙ radicals are generated by coincidental excitation in proximity to cold ones (that is, transient coexistence of two spatially close radicals only differing in their state populations), the electron transfer can occur. Therefore, the resulting SDET-induced net heterolysis offers an expedient and chemospecific means to edit the fission of symmetric and homopolar σ-bonds by means of state-directed poling of the incipient radical pair (that is, directional charge separation between state-differentiated,

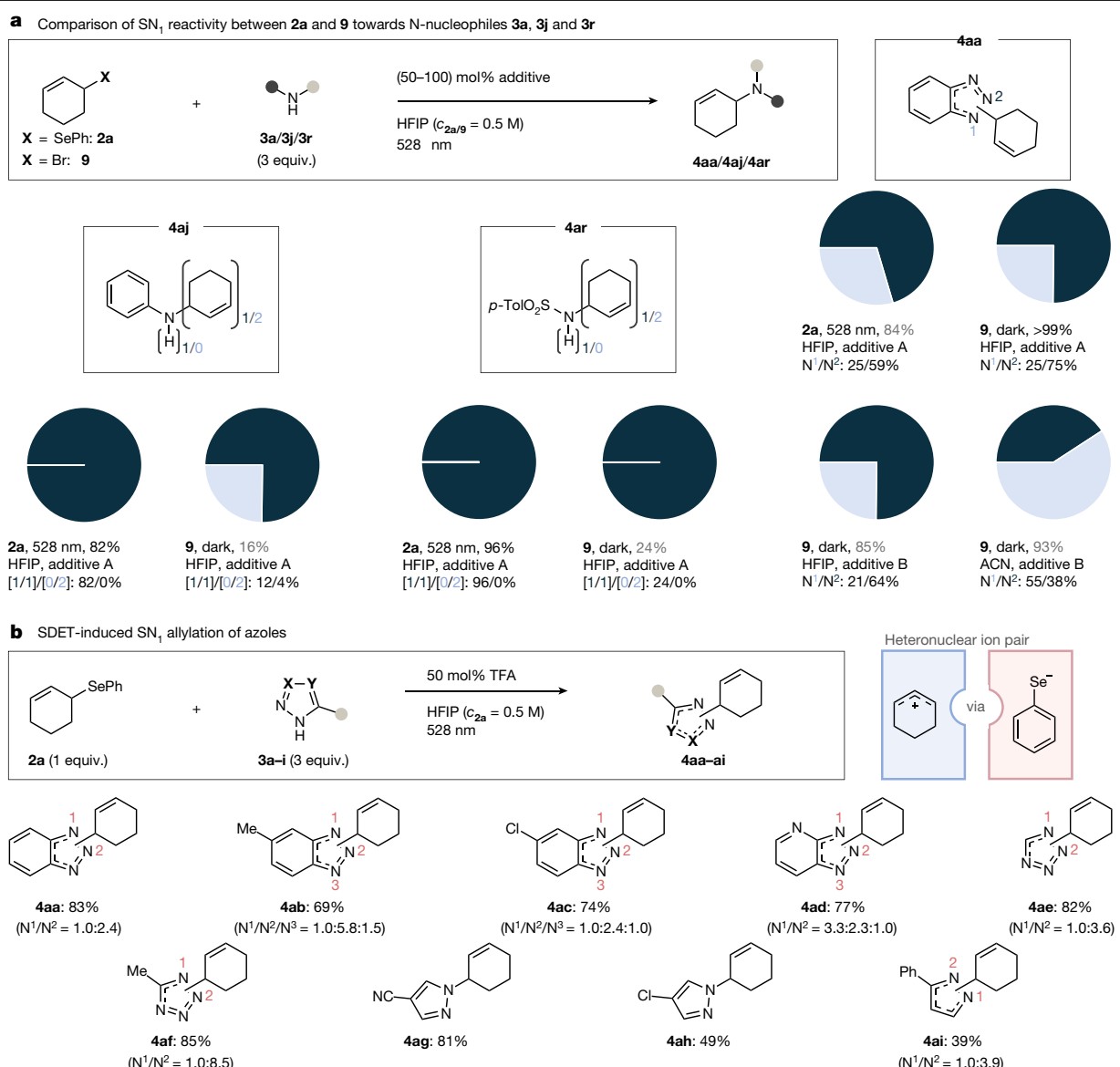

**Fig. 4 | Comparative reactivity study of 2a versus 9 and SDET-induced atypical $S_N1$ amination of allyl- and alkyl(aryl)selanes. a**, Reactivity comparison between **2a** and **9**. Pie charts indicate regioselectivity ratios of N[1] substitution (light blue) versus N[2] substitution (dark blue). Yields were determined by [1]H NMR spectroscopy using 1,3-dinitrobenzene as an internal standard. Additive A, 50% TFA; additive B, 1 equiv. $K_2CO_3$; ACN, acetonitrile. **b**, Exemplary series of SDET-induced allylations of azoles. Yields refer to isolated compounds.

former bond members). In total, this lends plausibility to the overthrowing idea that the SDET principle might indeed be applicable to other kinds of homopolar σ-bond constitutions as long as they are susceptible to similar two-step activations. It is important to point out that the observed requirement for keeping the state-differentiated radical pair members in close proximity to one another is most probably key to establish chemospecificity[31]. More concretely, provided that radical recombination before the SDET or back electron transfer and filial ion pair recombination after the SDET are the only unproductive processes kinetically feasible under said diffusion restrictions, unwanted detrimental side reactivity of the electrophile becomes improbable.

Next, we subjected diselane **1a** to 5-phenylpent-3-enoic acid (**6a**) (1.0 equiv.) in HFIP in the presence of TFA (0.5 equiv.) at 528 nm irradiation to test whether corresponding lactone **7a** is formed as a result of the anticipated 1,2-anti-addition of the SDET-generated PhSe[+] cation onto the alkene moiety (Extended Data Fig. 2a). To our delight, product **7a** was produced in 27% after 24 h. Changing to 447 nm irradiation gave clean access to lactone **7a** in an increased yield of 72% at 77% conversion.

Switching from TFA to methanesulfonic acid (MSA) led to quantitative product formation. In the absence of any acid additive, formation of product **7a** (61%) was still observed at full conversion alongside with butenolide **8a** (21% yield, entry 4). This outcome strongly supports our interpretation that the σ-bond ampholysis is not a consequence of Brønsted-acid catalysis. This notion is further corroborated by the fact that without light, but in the presence of either acid additive (TFA or MSA), no substantial product formation was recorded (entries 5 and 6; Supplementary Information, Chapter 7.6). We propose that the acid additive, together with the solvent, serves two purposes: (1) it kinetically assists in the Se–Se homolysis step (vide supra) and (2) it enhances the protonation of the PhSe[−] anion after the SDET event, thus precluding it from recombination with its counterion. Consequently, these data show that net heterolyses of symmetric σ-bonds are indeed possible through SDET activation.

Encouraged by the observation that butenolide **8a** can be directly accessed from enoic acid **6a**, we next tested whether the SDET principle could be further elaborated into a catalytic regime[32], in which

lactone **7a** only serves as an intermediate en route to butenolide **8a** (Extended Data Fig. 2b and Supplementary Tables 3–9). After minor modifications of the reaction conditions (405 nm, 35 °C), compound **8a** was obtained in 67% using only 5 mol% of diselane **1a**. This finding represents an example of a Lewis acid whose activity is reversibly elicited by a photochemical stimulus (that is, photo-Lewis acid)[33–37]. This new photo-Lewis-acid protocol also proved suitable for an exemplary series of other enoic acids **6b–f** (Extended Data Fig. 2b), which prefigures the potential that SDET-induced ampholyses might unfold in the realm of dinuclear catalysis[38–41].

To address question (2), we compared the electrophilic reactivity profile of SDET-activated allylselane **2a** with that of 3-bromocyclohex-1-ene (**9**) (Fig. 4a and Supplementary Information, Chapter 12). In HFIP, compounds **2a** and **9** showed similar outcomes with regard to crude yields (84 and >99%, respectively) and $N^2/N^1$ regioselectivities ($N^1$ in light blue, $N^2$ in dark blue). When electrophile **9** was reacted in acetonitrile in the presence of $K_2CO_3$ as a base, the yield remained high (85%) but the selectivity for $N^2$ allylation decreased to 41%, emphasizing the decisive impact of HFIP on the regioselectivity. Comparison of allyl(aryl)selane **2a** and bromide **9** in their reactivities towards less reactive N-nucleophiles revealed that the SDET activation provided superior yields in both tested cases, that is, aniline **3j** (82 versus 16%) and tosylsulfonamide **3r** (96 versus 24%, Fig. 4a and Supplementary Tables 24 and 25).

To ensure that the $S_N1$ reaction is driven only within the absorption window of PhSe⋅ (300 to 630 nm), the experiment was repeated with an optical filter setup that allowed irradiation within segments of an overall spectral range between 580 and 930 nm with varying LEDs (Supplementary Information and Supplementary Table 14, entries 1–8). Irradiation between 580 and 710 nm furnished product **4aa** in an unaltered yield of 84%. Changing the window to 630–710 nm (630 nm represents the endpoint of the absorption band of PhSe⋅) led to a substantial decrease in the conversion and yield to 8 and 6%, respectively (Supplementary Table 14, entry 6). Using an irradiation window from 700 to 810 nm only led to background reactivity (Supplementary Table 14, entry 4) with less than 5% conversion. These outcomes substantiate our hypothesis that the requisite cation **5a⁺** is indeed only generated in response to PhSe⋅ being excited with a suitable wavelength followed by a SDET. In addition, the key effect of HFIP relative to $^i$PrOH (and other solvents, Supplementary Table 11) was confirmed, as the yield dropped down to 10% in the latter solvent under otherwise unaltered conditions. Other azoles were also effective and provided target structures **4ab–ai** in isolated yields ranging from 39 to 85% (Fig. 4b).

Non-heteroaromatic N-nucleophiles such as hydrazides **3o–q**, anilines **3j–n**, and sulfonamides **3r–t** were also readily converted into their respective allylation products **4aj–at** (yield 31 to 90%, Extended Data Fig. 3). In addition to allylations (**4aa–cr**), the SDET protocol proved also effective for alkylations using alkylselanes **2d–f**, which provided access to secondary sulfonamides **4dr–fr** in isolated yields of up to 70%. Notably, substitution was even possible at the bridgehead carbon atom of bicyclo[2.2.2]octane **2f**, furnishing product **4fr** in 20% along with 20% of HFIP ether **4fr'** within 6 days reaction time. According to Bartlett and Knox[10], such $S_N1$ reactions are typically unfeasible under conventional $S_N$ conditions due to an increase in strain energy during the departure of the nucleofuge[9]. Single-step heterolyses at such positions are only known to be possible with very strong leaving groups such as $N_2^+$ (that is, strongly heteropolar C–N σ-bond)[42,43]. Our results indicate that this level of very high nucleofugality can indeed be emulated even with homopolar σ-bonds under SDET conditions, which is exemplarily further demonstrated in other $S_N1$ reactions typical to carbenium ions, such as Friedel–Crafts alkylations (Supplementary Information, Chapter 4.4 and Supplementary Scheme 3).

In summary, we have presented a comprehensive mechanistic and methodical study on the net heterolysis of symmetric and homopolar σ-bonds by means of SDET. Symmetric Se–Se single bonds (exemplified by diselane **1a**) can undergo stimulated ampholysis into Se-centred ion pairs; an outcome that has—to the best of our knowledge—never been shown with any kind of symmetric single bond. This finding complements long-standing notions on the general possibility of splitting symmetric and homopolar σ-bonds into (ambi)polar fragments in a unimolecular manner by sequential energy input (thermal or photochemical) by way of transient radical formation and subsequent differentiation of their electronic and/or vibrational state populations. The implications of our findings are in as far long-ranging as, in theory, the SDET-activation principle might be applicable to other pairs of equally electronegative nuclei (for example, bimetallic complexes[38–41], nitrogen–nitrogen[44,45], carbon–carbon[44,45]). This prospect may enable heretofore unexplored, inimitable reaction manifolds that are elusive to processes exclusively operating through thermal or photochemical single activation. The plausibility of this prospect was exemplified by (1) an atypical 1,2-anti-addition of diselane **1a** to enoic acid **6a** (Extended Data Fig. 2a), (2) the implementation of the ampholytic poling principle into catalytic manifolds (Extended Data Fig. 2b) and (3) highly chemospecific $S_N1$ reactions of allyl- and alkyl(aryl)selanes with N and C nucleophiles (Fig. 4b, Extended Data Fig. 3 and Supplementary Scheme 3), which in total underscores the unique synthetic utility of the SDET approach. Future investigations in our groups will focus on the implementation of the SDET concept in new catalytic transformations and its generalization onto other geminate, incipient radical species.

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

## Methods

### Absorption spectra of reaction constituents in the SDET-induced net heterolysis

We first characterized the anticipated selenium intermediates resulting from the proposed SDET-induced net heterolysis of diselane **1a** (that is, PhSe$^•$, PhSe$^-$ and PhSe$^+$, Extended Data Fig. 1a,b) in more detail. Investigations began with the elucidation of the spectral signatures of the expected intermediates and the identification of potential absorption overlaps under the reaction conditions between them or with any other reaction constituent (for example, nucleophile **3a** or product **4aa**, Extended Data Fig. 1). Ito and Tojo et al. reported a $\lambda_{max}$ for PhSe$^•$ of 450 and 490 nm, with a measurable absorption up to 630 nm (refs. 46,47). Furthermore, Lewis et al. documented two absorption maxima for PhSe$^-$ at 227 and 295 nm (ref. 48). To determine the wavelength at which the absorption of PhSe$^-$ potentially overlaps with that of PhSe$^•$, we generated NaSePh from (PhSe)$_2$ and NaBH$_4$. The corresponding absorption spectrum of PhSe$^-$ showed a signature only below 350 nm (Extended Data Fig. 1b, middle), thus confirming that co-excitation of both selenium species above 350 nm is impossible. Analogously, the spectral signatures of PhSe$^+$ (recorded from a mixture of *N*-phenylselanylphthalimide and MSA) range up to 500 nm (Extended Data Fig. 1b).

The unambiguity of our spectral assignment for the radical and ionic selenium species (Extended Data Fig. 1a,b) was also corroborated by computational means (Supplementary Figs. 22–24). The absorption spectra of PhSe$^•$ ($\lambda_{max}$ of 490 nm) and PhSe$^-$ ($\lambda_{max}$ of 295 and 227 nm in MeCN)[48] were reproduced by single reference computational methods (Supplementary Figs. 22 and 24) with error margins below 0.25 eV, which marks a high level of accuracy given the current state of the art for the simulation of such systems[49]. The spectral simulation of PhSe$^+$, on the other hand, was found to be complicated by the electronic configuration of this heavy ion, requiring multireference treatment to accurately reproduce the experimental spectrum computationally (Supplementary Information, Chapter 8). Considering the body of spectroscopic and computational data, it stands to reason that photo-excitation of PhSe$^•$ between 500 and 630 nm does not lead to substantial overlap with the observed absorption spectra of the presumed filial selenium ion pair (PhSe$^-$/PhSe$^+$) and of (PhSe)$_2$ itself, which does not exceed 430 nm (Extended Data Fig. 1b, bottom).

An analogous spectral analysis was also conducted for the cyclohexenyl intermediates **5a$^•$** and **5a$^+$** (that is, homopolar–heteronuclear case, Extended Data Fig. 1d), which were expected to result from the SDET-fragmentation of substrate **2a**. Schuler et al. reported radical **5a$^•$** to possesses a $\lambda_{max}$ of 240 nm with an absorption range of only up to 320 nm (ref. 50). Carbocation **5a$^+$**, generated by us in situ from 3-chlorocyclohex-1-ene in the presence of AgPF$_6$, shows similar spectral features as its radical congener, which do not exceed 300 nm (Extended Data Fig. 1d, top). In conclusion, both results confirm that—apart from PhSe$^•$—none of the reactants, solvent, additives (Extended Data Fig. 1c,d) and conceivably relevant reactive intermediates show a pronounced absorption at wavelengths greater than 500 nm.

### Data availability

Details on the procedures, optimization, characterization, and mechanisms, including spectra of new compounds and compounds made using the reported method, are available in the Supplementary Information.

### Code availability

In the paper, we used two ab initio quantum chemistry packages. The ORCA v.5.0.4 package is freely available (no costs apply, and the package can be downloaded following registration at https://www.orcaforum. kofo.mpg.de). The OpenMOLCAS package is an open-source software, published under the GNU LGPL license at https://gitlab.com/Molcas/ OpenMolcas. The TheoDORE software package is an open-source code and published under the GNU GPL license at https://github.com/ felixplasser/theodore-qc. All input files, postprocessing procedures and scripts used to evaluate theoretical data are available from the authors on request.

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

**Acknowledgements** The European Research Council (ERC Starting Grant 'ELDORADO' (grant agreement no. 803426) to A.B.) and the Bischöfliche Studienförderung–Cusanuswerk (PhD scholarship to A.F.T.) are acknowledged for financial support. Further, the project was funded by the Deutsche Forschungsgemeinschaft (DFG, the German Research Foundation) grant no. TRR 325–444632635 (projects B4 (to A.B. and C.H.N.), A5 (to P.N., D.J.G. and C.A.) and B5 (to J.R. and E.H.)). S. Bergwinkl is acknowledged for operational support during absorption spectroscopic measurement. A. Yelboga (substrates **2d**, **4ag**, **4an**, **4au**, **4br**), M. Seidl (substrates **1g**, **2f**), K. Müller (substrates **2a–1d**, **2e**) and K. Prosian (substrate **2b**), T. Lei (substrate **6d**, **6e**, **6f**) and F. Tannert (substrates **6b**, **6c**, **8a**, **8b**, **8c** and optimization reactions) are acknowledged for performing some of the substrate syntheses mentioned to their names. S. Mai is acknowledged for technical and operational support with the computational simulation of absorption spectra. Open access funding was provided by the ERC. We thank the late Prof. Josef Michl for his advice and inspiration to this work.

**Author contributions** J.R., P.N., A.B. conceived individual aspects of the idea. J.R. partially designed and managed the computational part, P.N. designed and managed the spectroscopic part, and A.B. designed and managed the synthetic part of the project. A.F.T. optimized the process, performed synthetic experiments, analysed the experimental data and prepared the Supplementary Information. D.J.G. and C.A. performed stationary and transient absorption spectroscopy on a µs timescale of model compounds and reactive intermediates and contributed to the writing of the Supplementary Information. E.H. and D.H.-C. conducted computational simulations of absorption spectra of reactive intermediates, proposed and guided by L.G. Further, E.H. and D.H.-C contributed to the computational thermodynamic analysis of stimulated and non-stimulated selenium and/or carbon doublet–doublet electron transfers, proposed and guided by L.G. and J.R. in complementary parts. J.R. proposed and guided mechanistic calculations of scrambling reactions and the analysis of the associated allylic selane frameworks, which were executed by E.H. Contributions to the Supplementary Information associated with all aforementioned computations were provided by E.H., D.H.-C., L.G., and J.R. C.H.N. optimized and performed the photo-Lewis-acid-catalysed lactonization reactions. R.J.K. performed transient absorption measurements of 1a on a fs to ns timescale and consulted in the interpretation of ultraviolet-visible light-spectroscopic data. P.R.N. documented the initial observation of a SDET-induced substitution of allylic selane 1a with nucleophiles, following an experimental design by A.B. K.Z. provided continuous consultancy during the project. All authors proofread the manuscript.

**Competing interests** The authors declare no competing interests.

**Additional information**
**Correspondence and requests for materials** should be addressed to Julia Rehbein, Patrick Nuernberger or Alexander Breder.

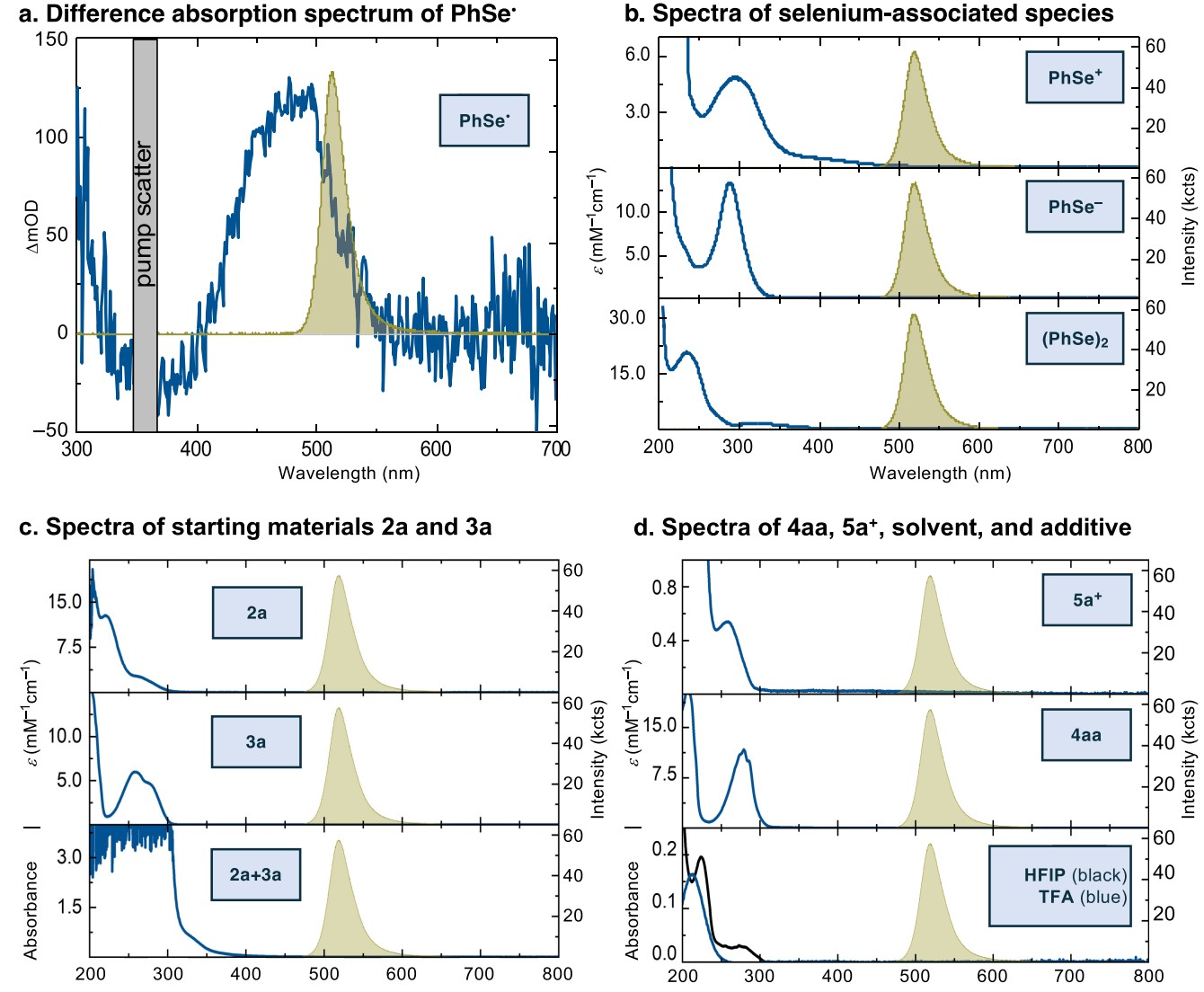

**a. Difference absorption spectrum of PhSe·**

**b. Spectra of selenium-associated species**

**c. Spectra of starting materials 2a and 3a**

**d. Spectra of 4aa, 5a⁺, solvent, and additive**

**Extended Data Fig. 1 | Absorption spectra of reaction constituents.**
**a**, Difference absorption spectrum of PhSe·. **b**, Spectra of PhSe⁺ (top),
PhSe⁻ (middle, taken in acetonitrile), and (PhSe)$_2$ (bottom). **c**, Spectra of
**2a** (top), benzotriazole (**3a**, middle), and a mixture of **2a** ($c$ = 11.1 mm),
**3a** ($c$ = 22.2 mm) (bottom). **d**, Spectra of **5a⁺** (top, taken in acetonitrile),

2-(cyclohex-2-en-1-yl)-2*H*-benzo[*d*][1,2,3]triazole (**4aa**, middle), and HFIP
(bottom, black curve) and TFA (bottom, blue curve). All spectra shown in **a** to **d**
were taken in HFIP, unless stated otherwise, and illustrate the LED emission
band (sketched in olive) used in the synthetic studies (Fig. 4 and Extended
Data Fig. 3).

## a. SDET-induced 1,2-addition of diselane 1a to enoic acid 6a

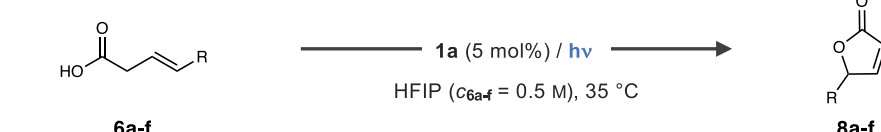

| entry | wavelength | solvent | additive | conversion [%] | 7a [%] | 8a [%] |
|---|---|---|---|---|---|---|
| 1 | 528 nm | HFIP | TFA | 61 | 27 | 0 |
| 2 | 447 nm | HFIP | TFA | 77 | 72 | 0 |
| 3 | 447 nm | HFIP | MSA | >99 | >99 | <1 |
| 4 | 447 nm | HFIP | none | >99 | 61 | 21 |
| 5 | no light | HFIP | TFA | 8 | 3 | 0 |
| 6 | no light | HFIP | MSA | 7 | 4 | 0 |
| 7 | 447 nm | $^i$PrOH | MSA | 59 | 47 | 0 |

## b. Application of the SDET-induced ampholysis in photo-Lewis-acid catalysis

1a (5 mol%) / hν

HFIP ($c_{6a\text{-}f}$ = 0.5 M), 35 °C

**6a-f** → **8a-f**

R:

Ph— | 4-MeO(C$_6$H$_4$)— | 4-F$_3$C(C$_6$H$_4$)— | 4-Cl(C$_6$H$_4$)— | 4-MeO$_2$C(C$_6$H$_4$)— | Cl—()$_4$

**8a**: 73% (67%) 24 h  **8b**: 51% (44%) 24 h  **8c**: 76% (71%) 24 h  **8d**: 63% (49%) 48 h  **8e**: 71% (61%) 48 h  **8f**: 80% (69%) 48 h

**Extended Data Fig. 2 | SDET-mediated 1,2-addition of 1a to 6a and design of the first photo-Lewis-acid catalysed lactonisation of alkenoic acids.** **a**, 5-phenylpent-3-enoic acid (**6a**, 1.0 equiv) was exposed to **1a** (1.0 equiv) and the corresponding acid additive (0 or 0.5 equiv) in HFIP or $^i$PrOH ($c_{1a}$ = 0.5 M) for 24 h at 19 °C and the indicated irradiation wavelength. **b**, Alk-3-enoic acids **6a-f** (0.5 mmol) were irradiated (405 nm) in the presence of **1a** (5 mol%) in HFIP ($c_{6a\text{-}f}$ = 0.5 M) for 24 h to 48 h at 35 °C. Yields in parentheses correspond to isolated compounds.

## *SDET-induced S$_N$1 allylation/alkylation of N-nucleophiles*

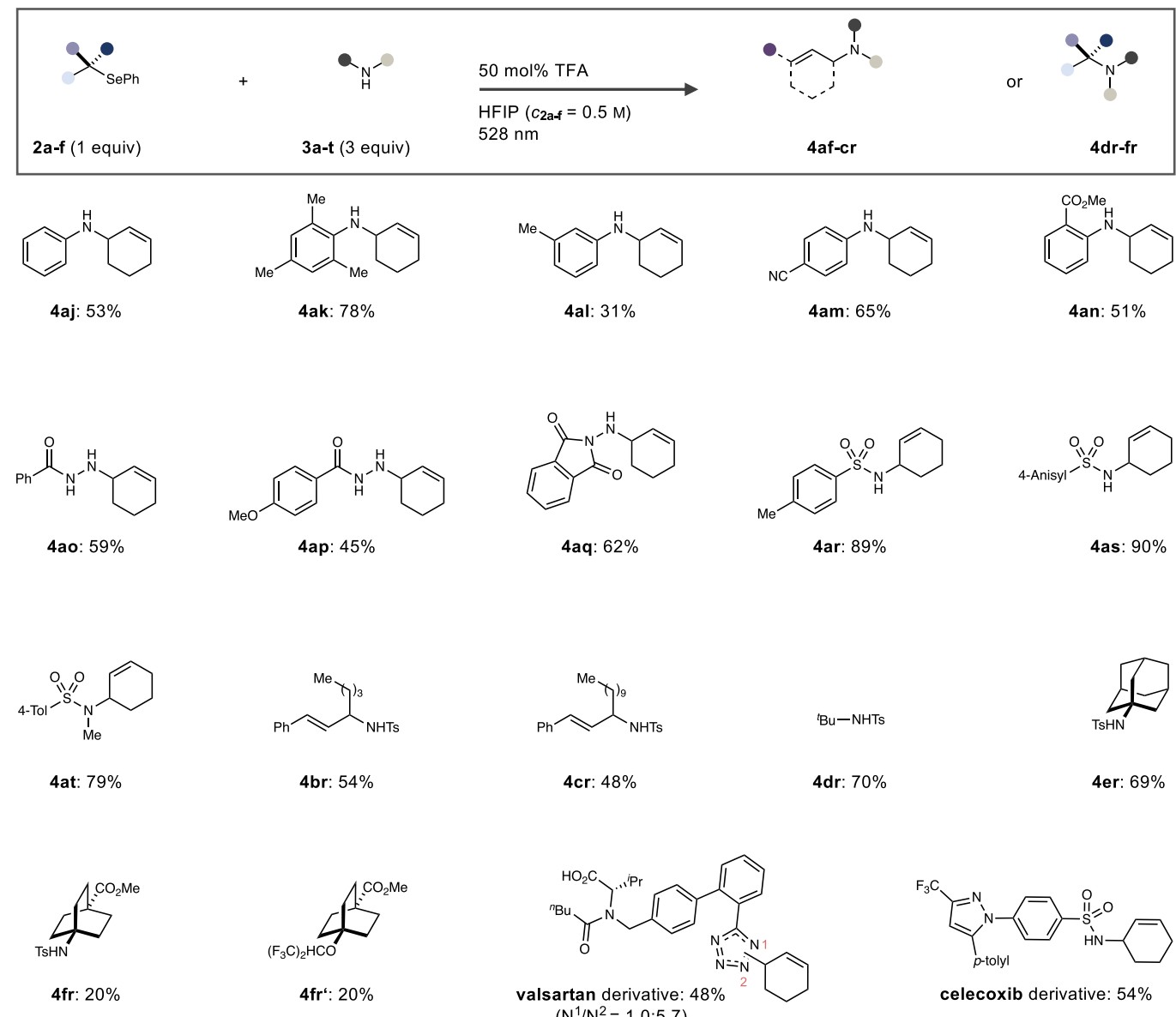

**Extended Data Fig. 3 | Exemplary series of SDET-induced allylations and alkylations of representative N-nucleophiles.** Yields refer to isolated compounds. [a]Selane **2f** was irradiated at 365 nm for 6 d under otherwise unaltered conditions to afford **4fr** and **4fr'** as an equimolar mixture. Yields of **4fr** and **4fr'** were determined by [1]H and [19]F NMR spectroscopy using 1,3-dinitrobenzene and benzotrifluoride, respectively, as internal standards.