## [Peer Review file · Nature]

Manuscript Title: Unimolecular net heterolysis of symmetric and homopolar σ -bonds

Reviewer Comments & Author Rebuttals

Reviewer Reports on the Initial Version:

Referees' comments:

Referee #1 (Remarks to the Author):

I am generally enthusiastic about this paper. Tiefel et al have proposed a novel solution to an old problem in organic chemistry relating to induced ionization of non-polarized bonds to generate reactive ionic intermediates. As the authors describe homo-polar or symmetric sigma bonds are highly biased toward homolytic pathways of bond cleavage, as the formal electron transfer that would convert the diradical pair to an ion pair is generally significantly endergonic. To overcome this to authors present a novel photochemical approach wherein initial photoinduced homolytic cleavage of a Se-Se or Se-C bond generates a radical pair. The incipient Se radical is then excited again via a second absorption event to yield an excited state that can engage in electron transfer with its geminate partner to yield an ion pair. The absorption properties of PhSe• above 500nm allow its selective excitation relative to other species and intermediates in solution, enabling the chemistry to proceed. In addition to spectroscopic evidence, synthetic applications to selenolactonization and amination reactions are presented.

I think this reactivity concept is very interesting and original, and it has the potential to be further extended into synthetically meaningful new technologies. The synthetic applications serve as proof of principle rather than addressing a meaningful problem, but I think that this is appropriate given the conceptual and physical emphasis of the paper. The key hypotheses are supported by the data presented and I believe the preponderance of evidence suggests that the postulated mechanism is operative. The paper is well written and SI is nicely put together and thorough. Altogether, I think the paper is suitable for Nature on grounds of novelty, cross-disciplinary interest, and potential impact, but there are a few small issues I hope the authors might clarify:

- What are the quantum yields for lactonization and amination reactions? Additionally, these are two-photon processes. Is a quadratic dependence on the photon flux observed?
- Related to the question above - how problematic is ion pair recombination (or back ET to reform the diradical pair)? If the symmetry breaking ET must occur in cage and there is a significant driving force to return the starting material, does this limit the efficiency of the process? Can these kinetics be extracted from the TA data on hand?
- Is there an estimate for the lifetime of the excited Se radical? Typically, doublet excited states are very short lived (sub nanosecond) – does this require the other radical species to be in cage for ET to occur? Is this a potential limitation? Some discussion of this point would be appreciated.

- How did the authors estimate the redox potential of the PhSe• radical in its excited state? Thermochemistry is presented in the SI but some comment
- What is the electronic structure of the PhSe• radical in its excited state? Some further description of this key intermediate would be nice to include.
- In table S11 why is there such a large difference between the BDE and BDFE for the C-Se bonds? This seems larger than usual difference for organic compounds.
- Related to this point, how do authors account for Se exchange in the dark after a few hours given the calculated BDFEs? Taking these values as the minimum activation energies suggests that thermal homolysis should be much slower. Are these chain mechanisms that only require initiation? Some comment here would be helpful.
- Can the authors provide more narrative descriptions and a physical justification for how HFIP stabilizes the Se radical intermediates?
- Since the cation has additional energetically favorable pathways for consumption via recombination or Back ET, is the strength of the nucleophile a meaningful parameter? In terms of the results in Figure 5, I wonder if weaker pi nucleophiles such as allylsilanes or electron rich aromatics are viable nucleophiles as well in addition to the heteroatomic nucleophiles shown?
- I appreciate that this is not a synthetically focused paper but I wonder if just a few additional carbon-based electrophiles could be included? I think this would magnify the impact of the synthetic sections.

Referee #2 (Remarks to the Author):

The manuscript describes a transformation in which homolytic thermolysis, followed by absorption of light, results in intermolecular electron transfer - overall resulting in the effective heterolytic cleavage of symmetric sigma bonds into cation/anion pairs. I find this concept stimulating, and the explanation put forward to be overall convincing. Without commenting on synthetic utility or novelty (which are outside of my area), I did have some comments about the supporting theoretical work used to interrogate the mechanism of the transformation:

Overall, the suite of methods used here is very comprehensive, including DFT, TD-DFT and multi-reference methods where appropriate. These calculations are used to support spectroscopic interpretation, and to make quantitative arguments about the favorability of SET in the ground vs. photochemically excited state. They largely succeed in this goal - but perhaps the following could be clarified/considered:

- the addition of acid additives like TFA have an effect on conversion. Did the authors consider

coordination of TFA to the selenium - is it possible to theoretically show that thermal heterolytic Se-Se cleavage is still relatively unfavorable?

- How reasonable is it to compare the computed absorption spectra of the "naked" PhS radical, anion and cation? It seems very likely that H-bonding to HFIP or TFA would influence the absorption of the anion in particular.

- the extended discussion on the multireference nature of the PhSe⁺ cation is interesting, but perhaps could be summarized in a sentence in the manuscript and the longer discussion moved to SI.

- The manuscript implies that the thermochemistry for electron transfer were computed using theoretically predicted reduction potentials for the SePh radical in ground and first excited state, in combination with an experimental value for oxidation of the allylic radical. Is there a good reason for this beyond availability of data? It would be very helpful to show that the fully theory-predicted values of DG give similar results.

- On the same topic, in computing the DG values for the formation of the cation/anion pair from two neutral, radical species - is there an implicit assumption of infinite separation here? In other words, is the favorable electrostatic interaction between cation and anion factored into the two DG values in Figure 3?

- Naive question, but is the terminology of a "doublet pair" standard? Each radical is a doublet, but overall the pair must adopt a singlet or triplet electronic state.

Author Rebuttals to Initial Comments

To Referee 1:

Q1: *What are the quantum yields for lactonization and amination reactions? Additionally, these are two-photon processes. Is a quadratic dependence on the photon flux observed?*

R1: Because of the very low concentration of the phenylselenenyl radical and the resulting very weak absorption, which is below the detection limit of our stationary UV/Vis spectroscopy, the quantum yield cannot be determined accurately. Instead, we aimed to obtain an estimate, for which two approximations are made, yielding a result whose order of magnitude is consistent with the proposed mechanism. The corresponding evaluation was added to Section 5 of the SI. In short, we assumed the radical concentration to be constant over the reaction time, given that both in the amination as well as in the lactonization reaction, diphenyl diselenide is present either as a byproduct or as a catalyst and is in equilibrium with the phenylselenenyl radical. Additionally, we expected the radical concentration to be at about 0.4 μM or below, which is too low for our spectroscopy to detect. With the additional knowledge of the emission power and spectrum of the LED, the reaction times and the yields, we were able to obtain an estimate for the quantum yields. For the amination reaction, this value was about 4 in presence of oxygen and about 1 in its absence, in line with our control experiments suggesting a conversion of up to 4 molecules per absorbed photon under air due to stoichiometric formation of hydrogen peroxide, as shown in Section 4.3 of the SI. In the case of the lactonization reaction, we estimated a quantum yield of 0.7 under air. Although hampered by non-negligible imprecisions because of the approximations made, and also taking some uncertainties in terms of minor background reactivity in the dark as well as potential weighing errors regarding the NMR-standard into consideration, the estimates for the quantum yields are compatible with our proposed mechanism.

Regarding the possibility of two-photon processes and consequently quadratic rate dependence in photons, we performed experiments with attenuating optical filters to systematically vary the photon flux into the reaction vessel. For both amination and lactonization in the absence and presence of oxygen, the conversion and the product yields exhibit a linear dependence on the photon flux. The corresponding data is now presented in Figures S5 and S6 (Section 5) in the SI. This outcome is in line with our expectations, as the first stimulus was anticipated to be a thermal homolysis, followed by a photonic excitation of the transient radicals. This mechanistic notion is consistent with the observed linear photon flux correlation.

Q2: *Related to the question above - how problematic is ion pair recombination (or back ET to reform the diradical pair)? If the symmetry breaking ET must occur in cage and there is a significant driving force to return the starting material, does this limit the efficiency of the process? Can these kinetics be extracted from the TA data on hand?*

R2: In general, we believe that the solvent plays a key role in this process, in part, for equivalent reasons as in regular S_N1 reactions. We think that common factors such as electrostatic and H-bonding interactions are operative in our reactions as well.

Regarding the efficiency of SDET-induced substitutions, we performed a series of competition experiments with allylic halides to obtain a quantitative estimate on the efficiency of the reaction, which is now summarized in Fig. 5a and Section 12 of the SI. We found that when *p*-tosylsulfonamide was used as a reference nucleophile and exposed to cyclohex-2-en-1-ylhalides (Hal = Cl, Br, Table S25) as well as to **2a** under identical conditions (HFIP, 528 nm, 50 mol% TFA, 19 °C, C_{electrophile} = 0.5 M), the S_N1 reaction of **2a** was more than twice as fast as cyclohex-2-en-1-ylbromide (bromide: 25% yield/83% conv. vs. **2a**: 58% yield/60% conv., 30 min.), and 10% faster than the corresponding chloride (52% yield/100% conv., 30 min.). Within 3 h, the reaction of **2a** reached 96% yield at full conversion whereas the bromide electrophile remained at a yield of 24%, but also reached 100% conversion. As we speculated that the latter result is a mere consequence of rapid degradation of the bromide, we added K₂CO₃ as base instead of the acid additive. This, indeed, increased the yield to 68% at full conversion after 3 h, but this outcome was still inferior by 30%. These findings already underscore that the SDET activation principle can generate carbon electrophiles that, in part, can compete with – or even outperform – the reactivity of common halide analogues. We rationalize the role of the solvent and acid additive in the following way: right after the SDET, the incipient selenide anion is intercepted through protonation by the pre-existing H-bonding network of the solvent/additive combination.

Q3: *Is there an estimate for the lifetime of the excited Se radical? Typically, doublet excited states are very short lived (sub nanosecond) – does this require the other radical species to be in cage for ET to occur? Is this a potential limitation? Some discussion of this point would be appreciated.*

R3: The computationally estimated excited state potential energy surfaces (Fig. S28) suggest a substantial activation barrier for the non-radiative deactivation of a cooled D₂ state to the D₁ state. Hence, depending on the rate of vibrational cooling via energy dissipation into the solvent, the excited radical may get stuck in the D₂ state, thus increasing its excited state lifetime. Our calculations lend plausibility to the assumption

that at least part of the chemical reactivity of the PhSe[•] radical may originate from a longevous non-Kasha state.

Despite this hypothesis, we cannot determine the lifetime of the electronically excited state of the Se radical from our spectroscopic data directly. As the reviewer rightfully points out, D_n states (n>0) typically show very short excited-state lifetimes. But this particular feature is believed to be an advantage for the following two reasons. Reason A) As we show in Fig. 4a, the SDET proceeds even between a vibrationally excited and a cooled PhSe[•] radical. The same might be operative in the case of the SDET-induced fission of carbon–selenium bonds, since we observe the same chemical behavior – i.e., ion pair formation. These findings suggest that excited state lifetimes are – at least as far as we have explored this kind of reactivity – not limiting the SDET. Reason B) Our results suggest that encounters of cold radicals only lead to radical recombination but not to SDET-induced ion pair formation. From this circumstance it follows that only spatially proximate (e.g., solvent-caged) and quantum state-differentiated (i.e., electronically or vibrationally) radical pairs can undergo the SDET. This means that most likely only radicals derived from the same bond can result in the ionization, and that this feature results in a very high chemospecificity for this kind of polar reactivity. We have now emphasized this aspect more strongly on p. 11 in the main text.

Q4: *How did the authors estimate the redox potential of the PhSe[•] radical in its excited state? Thermochemistry is presented in the SI but some comment [would be helpful].*

R4: To estimate the reduction potential of the selenyl radical in its excited state, the Gibbs free energy was needed. Hence, we performed a TDDFT geometry optimization in the D₁ and D₂ state using equilibrium CPCM solvation followed by a numerical frequency calculation to estimate thermodynamical contributions to the energy (see updated SI, Section 9). Following the reviewer's comment, we have now updated SI, Section 9, with the corresponding equations and explanations that illustrate the respective details of these calculations.

Q5: *What is the electronic structure of the PhSe[•] radical in its excited state? Some further description of this key intermediate would be nice to include.*

R5: Irradiation with the experimentally used wavelengths mainly excites the bright D₃ state of π-p excitation character. Furthermore, the D₂ state is of π-p excitation character however with larger charge transfer contributions explaining the low extinction coefficient. The energetically lowest electronically excited doublet state (D₁) holds a n_{Se}-p charge transfer character resulting in a shift of the spin density to the selenium p-orbital which is orthogonally oriented to the π-system of the phenyl ring. We thank the reviewer for pointing this out aspect, and have now included an extensive discussion about these excited states in the SI (see new SI, Section 8.3).

Q6: *In table S11 why is there such a large difference between the BDE and BDFE for the C-Se bonds? This seems larger than usual difference for organic compounds.*

R6: Considering the large differences between BDE and BDFE, entropic contributions have to play a major role in the dissociation of the allylselane. Hence, we investigated the contributions to these entropic terms in more detail and identified the rotational entropy of the allylselane as the main contributor. A relaxed potential energy surface scan showed that the torsion around the C–Se bond, connecting the allyl-fragment with

the selenium atom, is kinetically hindered, exhibiting three distinct minima. We have now included this analysis and the corresponding surface scan in the SI, Section 10.2.

Q7: *Related to this point, how do authors account for Se exchange in the dark after a few hours given the calculated BDFEs? Taking these values as the minimum activation energies suggests that thermal homolysis should be much slower. Are these chain mechanisms that only require initiation? Some comment here would be helpful.*

R7: Indeed, Se–Se bonds as well as Se–C bonds had been identified in the past to belong to the dynamic covalent bonds. Their BDEs between 40-50 kcal/mol had been used to rationalize their aptitude to exchange with other Se–C and Se–Se units in a bimolecular fashion (see revised SI for updated information and citations) and this behavior has found its application in self-healing polymers for instance. Nevertheless, it is quite apparent that under the assumption of a LFER the values of the BDEs/ BDFEs do not suggest that there is a quick scrambling possible at room temperature. The answer is therefore yes, a probable mechanism is – as outlined in the SI – a two-step process: initiation, i.e. formation of the first Se-centered radical followed by a bimolecular scrambling between Se-radical and starting material. This mechanism captures the kinetics sufficiently well and explains the experimentally observed scrambling. The formation of this very initial Se-centered radical from a homolysis of the Se–Se or Se–C bond however is hard to predict with the given means of QM calculations preventing the inclusion of a full 1st and 2nd solvation shell. The partial explicit solvation by a trimer of HFIP did not significantly lower the BDE/BDFE values, indicating that the solvation effect must go beyond this simple H-bonding network. Therefore, this aspect remains a critical questions to answer by computations. With regard to the current study, we chose to focus on the elucidation of SDET mechanism with highest priority. It should be pointed out, however, that we did not observe in our control experiments any enhancement in scrambling rate when adding diphenyldiselane as a potential initiator to the scrambling experiments of the allylselanes. These findings suggest that traces of diphenyldiselane are most likely not the source for initial radical formation, which renders the direct thermal homolysis of allylselanes still the likeliest scenario based on our current knowledge.

Q8: *Can the authors provide more narrative descriptions and a physical justification for how HFIP stabilizes the Se radical intermediates?*

R8: To a first approximation, there are two critical factors to be considered, which we are very interested to distinguish between. More concretely, a) HFIP stabilizes the Se-radicals thermodynamically by explicit solvation or b) it renders the radicals kinetically inert by changing their electronics because of H-bond interactions. Based on the stabilization energies and the FMOs, the ‘HFIP-effect’ seems to be best described as a kinetic effect. The computed energies of the alpha-spin singly occupied molecular orbitals of the PhSe[•] radical show strong dependence on explicit solvation. This energy decreases going from implicit solvation to explicit solvation with 2-propanol to explicit solvation with HFIP. Hence, chemical loss paths such as radical recombination will get kinetically hindered. We included these calculated orbital energies and the corresponding discussion in the SI, Section 10.3.

Q9: *Since the cation has additional energetically favorable pathways for consumption via recombination or Back ET, is the strength of the nucleophile a meaningful parameter? In terms of the results in Figure 5, I wonder if weaker pi nucleophiles such as*

allylsilanes or electron rich aromatics are viable nucleophiles as well in addition to the heteroatomic nucleophiles shown?

R9: We would like to address this question in three parts. 1) Looking at the Mayr nucleophilicity scales (tabulated at: <https://www.cup.lmu.de/oc/mayr/reaktionsdatenbank/>), the examples shown in Fig. 5 cover a range of about 14 units of magnitude with HFIP ($N = -1.93$, $s_N = 1.09$) being the weakest nucleophile (cmpd. **4fr'**) and aniline ($N = 12.64$, $s_N = 0.68$) the strongest one being tabulated (cmpd. **4aj**). Some of our nucleophiles might even be stronger, but we could not find the corresponding N parameter. Since we used the cyclohexenyl cation as the reference electrophile in HFIP as the reference solvent for most of our nucleophiles, we argue that the Mayr nucleophilicity parameters can be regarded at least as qualitatively indicative for the broad tolerance toward the nature of the nucleophiles, which leads to the second part of our answer.

2) For the cyclohexenyl cation we could not find the corresponding E parameter in the Mayr tables. However, we looked up the hydride ion affinity values instead (*J. Am. Chem. Soc.*, **101**, 1239 (1979); *J. Am. Chem. Soc.*, **101**, 4067, (1979)). We are aware that these values are recorded in the gas phase, which makes them for certain less corroborative for our conditions. Nonetheless, the tabulated data show that cations with HIA values as low as 225 kcal/mol (cyclopentenyl cation as the closest homolog we could find data for). By now, we have tested electrophiles that we suspect of being either just as weak as the cyclopentenyl cation or even weaker such as the methoxymethyl cation (not included in this study, but tested in other reactions performed in the meantime). In conclusion, we posit that BET and ion recombination do not play a hampering role in our title procedure, as we could show that the reaction provides about 60% product and 87% conversion within 20 minutes. As is emphasized in **R2**, the solvent/additive combination most likely captures the transient anion through protonation at a superior rate, thus effectively preventing the title process from both unproductive pathways.

3) We could show that π -nucleophiles also work under our conditions. We included an exemplary series of SDET-induced Friedel-Crafts reactions in the SI (Scheme S3, Section 4.4). Although the yields are, in part, moderate due to difficulties during the separation of nonpolar diselenide byproducts from the allylated and alkylated arene products, the data unambiguously shows that carbon nucleophiles are in fact competent reaction partners.

Q10: *I appreciate that this is not a synthetically focused paper but I wonder if just a few additional carbon-based electrophiles could be included? I think this would magnify the impact of the synthetic sections.*

R10: We agree with the assessment of referee 1, and included a few more synthetic examples. On the one hand, we included a catalytic version of the lactonization – now to be found in Fig. 4c. We think that these additional examples unequivocally showcase that the SDET-induced amphotolysis of symmetric σ -bonds can be implemented in synthetic contexts for which there are no equivalent examples known in the literature. More concretely, the chemistry of photoacids and photoacid generators has become increasingly popular outside the realm of material sciences. While for the class of photoacid generators (i.e., compounds being capable of irreversibly releasing an acid upon photoexcitation) both Lewis- and Brønsted-acid compounds are known, there are only examples of Brønsted-photoacids (i.e., compounds that alter their acidity for the duration of photoexcitation) on record so far. In analogy to the concept of “metastable state Brønsted-photoacids”, first reported by Liao et al. (*J. Am. Chem. Soc.*, **133**,

14699–14703 (2011)), we now show that SDET-released PhSe^+ cations act as potent photo-Lewis-acid catalysts that can be exploited for the electrophilic activation of π -bonds (Fig. 4c). On the other hand, we followed the suggestion of referee 1 and tested a representative series of C-nucleophiles (Scheme S3). As was anticipated by the referee, electron rich arenes readily undergo SDET-induced Friedel-Crafts chemistry (see also answer R9).

To Referee 2:

Q1: *the addition of acid additives like TFA have an effect on conversion. Did the authors consider coordination of TFA to the selenium - is it possible to theoretically show that thermal heterolytic Se-Se cleavage is still relatively unfavorable?*

R1: We addressed this point experimentally. In HFIP, the addition of 107 mM methylsulfonic acid (MSA) indeed leads to heterolytic cleavage of the Se–Se bond in $(\text{PhSe})_2$ and the formation of PhSe^+ , which can be monitored by spectroscopy (Fig. S19 in the SI). However, the formation of PhSe^+ takes place on a timescale of hours, while our reactions are completed within minutes. Furthermore, it needs to be stated that a 738-fold excess of MSA was used for this measurement.

We also looked into trifluoroacetic acid (TFA, SI, Fig. S20 and S21), which is a significantly weaker acid than MSA. In a 0.3 M concentration of TFA in HFIP, which is slightly higher than the concentration used in our title conditions, there was no sign of $(\text{PhSe})_2$ being converted into PhSe^+ within a time span of 15 hours. Increasing the concentrations to 2.6 M TFA, which is a 10-fold increase in concentration compared to our title conditions, a very slow reaction can be observed. Based on these observations, we consider a kinetically relevant contribution of TFA to the formation of the PhSe^+ cation in the dark as being ruled out. The dark reaction is significantly slower than the reaction under light. However, the acid mediated PhSe^+ might explain the background reaction observed in the dark.

Q2: *How reasonable is it to compare the computed absorption spectra of the "naked" PhSe radical, anion and cation? It seems very likely that H-bonding to HFIP or TFA would influence the absorption of the anion in particular.*

R2: Since the cation is positively charged and complex in its electronic nature, explicit solvation effects were neglected. However, for both the phenyl selenyl anion and radical we calculated the TDA spectra including explicit solvation by one HFIP trimer coordinating to the selenium atom (Fig. S22, S24). In both cases, the shifting in absorption wavelengths were only of minor nature. While the anion was shifted even more to the blue, the radical was shifted slightly to the red, even improving the match to the experimental spectrum.

Q3: *the extended discussion on the multireference nature of the PhSe+ cation is interesting, but perhaps could be summarized in a sentence in the manuscript and the longer discussion moved to SI.*

R3: We thank the reviewer for this suggestion, and have now modified the text accordingly.

Q4: *The manuscript implies that the thermochemistry for electron transfer were computed using theoretically predicted reduction potentials for the SePh radical in ground*

and first excited state, in combination with an experimental value for oxidation of the allylic radical. Is there a good reason for this beyond availability of data? It would be very helpful to show that the fully theory-predicted values of DG give similar results.

R4: All mentioned redox potentials were computationally estimated. We revised the wording in the manuscript to circumvent misunderstandings.

Q5: *On the same topic, in computing the DG values for the formation of the cation/anion pair from two neutral, radical species - is there an implicit assumption of infinite separation here? In other words, is the favorable electrostatic interaction between cation and anion factored into the two DG values in Figure 3?*

R5: We indeed assumed infinite separation between both species. We have now provided a more complete explanation in the SI, Section 9.2, where we attempt an estimate of an upper barrier of the absolute Coulombic interaction. The value of this barrier for the attractive Coulombic interaction is slightly greater than the exergonicity of the electron transfer from the D₁ state, while it is much lower than the exergonicity of the electron transfer from the D₂ state. The difference in Gibbs free energy for the electron transfer from the D₁ state has to be therefore considered an upper limit, as we now discuss in the SI, Section 9.2.

Q6: *Naive question, but is the terminology of a "doublet pair" standard? Each radical is a doublet, but overall, the pair must adopt a singlet or triplet electronic state.*

R6: We believe that the question is absolutely justified. The two instances in which the term “doublet pair” has remained unintentionally in the manuscript in Fig. 1 and 3 represent artefacts, and should have been (and are now) replaced with the term “radical pair”. Nonetheless, we would like to emphasize that our terminology used for the key electron transfer itself, i.e., stimulated doublet-doublet electron transfer, falls into the same terminology commonly used to describe triplet-triplet annihilations (via Dexter electron transfer) and singlet fissions (e.g., *J. Phys. Chem. A*, **119**, 12699–12705 (2015); *Energy Environ. Sci.*, **15**, 4982–5016 (2022); *Phys. Rev. B: Condens. Matter Mater. Phys.*, **1**, 896–902 (1970)). In all of these publications, the term “triplet-triplet pair” is used despite the fact that the global multiplicity of the spin-correlated systems equals a singlet or a quintet state. Given the mechanistic similarity of these processes and ours, we find that the term “doublet-doublet electron transfer” is just a logic and direct adaptation of existing terminology.

Reviewer Reports on the First Revision:

Referees' comments:

Referee #1 (Remarks to the Author):

All of my questions have been satisfactorily addressed. I am happy to support publication of this work.

Referee #2 (Remarks to the Author):

The authors have responded in detail to the reviewers' comments.

The SI addresses these technical questions with greater clarity on theoretical protocols, and the expanded discussion there is appreciated. In particular the expanded discussion on the doublet state excited state PES in D1,D2 and D3 is particularly insightful and adds to the mechanistic interest overall.

Overall my enthusiasm for the high-level concept and the quality of the work remains high, and I have no additional technical queries.